# Mental health and help seeking among trauma-exposed emergency service staff: a qualitative evidence synthesis

Niklas Maximilian Auth ![ORCID],[1] Matthew James Booker,[2] Jennifer Wild ![ORCID] ,[3] Ruth Riley ![ORCID] [4,5]

¹College of Medical and Dental Sciences, University of Birmingham, Birmingham, UK
²School of Social and Community Medicine, University of Bristol, Bristol, UK
³Experimental Psychology, University of Oxford, Oxford, UK
⁴Social and Community Medicine, University of Bristol, Bristol, UK
⁵Department of Applied Health Research, University of Birmingham, Birmingham, UK

**Correspondence to**
Niklas Maximilian Auth;
nma676@student.bham.ac.uk

## ABSTRACT

**Objectives** To identify factors and contexts that may contribute to mental health and recovery from psychological difficulties for emergency service workers (ESWs) exposed to occupational trauma, and barriers and facilitators to help-seeking behaviour among trauma-exposed ESWs.

**Background** ESWs are at greater risk of stressor-related psychopathology than the general population. Exposure to occupational stressors and trauma contribute to the observed rates of post-trauma psychopathology in this occupational group with implications for workforce sustainability. Types of organisational interventions offered to trauma-exposed ESWs are inconsistent across the UK, with uncertainty around how to engage staff.

**Design** Four databases (OVID MEDLINE, EMBASE, PsycINFO and SCOPUS) were systematically searched from 1 January 1980 to March 2020, with citation tracking and reference chaining. A modified Critical Appraisal Skills Programme tool and quality appraisal prompts were used to identify fatally flawed studies. Qualitative studies of trauma-exposure in front-line ESWs were included, and data were extracted using a customised extraction table. Included studies were analysed using thematic synthesis.

**Results** A qualitative evidence synthesis was conducted with 24 qualitative studies meeting inclusion criteria, as defined by the PerSPEcTiF framework. Fourteen descriptive themes emerged from this review, categorised into two overarching constructs: (1) factors contributing to mental health (such as the need for downtime, peer support and reassurance) and (2) factors influencing help-seeking behaviour (such as stigma, the content/form/mandatory nature of interventions, and mental health literacy issues including emotional awareness and education).

**Conclusion** ESWs reported disconnect between the organisations' cultural positioning on trauma-related mental health, the reality of undertaking the role and the perceived applicability and usefulness of trauma interventions. Following traumatic exposure, ESWs identify benefitting from recovery time and informal support from trusted colleagues. A culture which encourages help seeking and open dialogue around mental health may reduce stigma and improve recovery from mental ill health associated with trauma exposure.

### Strengths and limitations of this study

► This is the first qualitative evidence synthesis of traumatisation and mental health help-seeking in emergency service workers (ESWs).

► A user advisory group of ambulance management stakeholders and ESWs was involved in the design and purpose of this research.

► Findings are drawn from pre-Covid literature, however, core themes are omnirelevant.

► Study quality varied significantly and there is a predominance of ambulance service literature in included studies.

## INTRODUCTION

Emergency service workers (ESWs) consisting of members of the emergency medical services (EMS), the fire service and the police force, consistently experience poorer mental health outcomes when compared with the general population. While subtle differences exist between occupational groups within emergency service organisations (ESOs), ESWs, also referred to as first responders, experience disproportionately higher rates of post-traumatic stress disorder (PTSD), anxiety, depression and psychological distress.[1–5]

Suicide attempts by ESWs are considerably more prevalent than the estimated rate of 0.5% in the general population.[6] Exposure to traumatic events accounts for higher PTSD rates in the ESW cohort,[2] and is the second most commonly reported cause of poor mental health among UK ESWs in a 2019 survey, following excessive workload.[7] This review will consider the terms 'traumatic incident/event' synonymously with the term 'critical incident', which is defined as: 'any event with sufficient impact to produce significant emotional reactions in people now or later', as described by Mitchell and Everly.[8] The increased incidence of PTSD in ESW populations[1 2] is important to appreciate as PTSD is a risk factor for suicidal ideation and

risky behaviours in civilian and military populations,[9] [10] and increases suicidal risk in ESWs.[11] [12] The wider impacts of mental ill health among ESWs include high rates of absenteeism and presenteeism, resulting in significant costs to ESOs.[13]

ESOs employ a variety of programmes to prevent the development of mental ill health in trauma-exposed staff, of which main categories include stress management, psychotherapy and health promotion.[14] Single session critical incident stress debriefing (CISD) following trauma exposure is a psychological intervention with widespread historical and current use in ESOs.[15–17] However, by the early 2000s, a substantial body of evidence demonstrated that CISD was neutral at best and harmful at worst with respect to preventing PTSD; it appears that CISD interferes with natural recovery.[15] [16] [18–20] The following two interventions are emerging among UK ESOs, and are examples of prevention strategies hoped to replace traditional debriefing methods. Trauma Risk Management (TRiM) is a peer support system, delivered by trained volunteers within the organisation,[21] who assess trauma exposed individuals for risk of mental ill health.[22] [23] The evidence surrounding TRiM's impact on users' mental health outcomes or for improving attitudes to mental ill health is inconclusive.[23–26] Schwartz rounds, best described as a cultural change initiative, also emerging in the UK,[27] allow multidisciplinary healthcare staff to share and discuss non-clinical aspects of their work, such as psychosocial, ethical and emotional issues.[28] As of May 2020, Schwartz rounds have been adopted by four UK ambulance trusts.[27] Schwartz rounds are reported to improve staff psychological well-being and increase 'empathy and compassion for colleagues',[29] although they have not been formally evaluated. Other interventions provided by UK ESOs to support the emotional well-being of ESWs after attending to critical incidents include counselling and 24-hour helplines,[30–34] 'defusing' programmes[35] which require staff who attended a critical incident to discuss facts surrounding the event in a structured group format, and peer support networks.[36]

Despite the availability of interventions, ESWs experience barriers to mental healthcare with one-third reporting that they experience mental health stigma, a rate that is higher than the general population.[37] The purpose of this review is to identify factors and contexts that may contribute to mental health and recovery from psychological difficulties experienced by ESWs exposed to occupational trauma. We were also interested in identifying barriers and facilitators to help-seeking behaviour among trauma-exposed ESWs.

When discussing methods aimed at improving mental health, 'protection' will be used in the context of trauma-exposed ESWs, while 'recovery' relates to ESWs experiencing psychological distress as a result of trauma-exposure. By identifying important contextual factors which help and hinder staff when they access support, and illuminating benefits and drawbacks of current organisational interventions, this review aims to offer qualitative insights grounded in the perceptions of ESWs, which may help ESOs in decision making about psychological support for their staff following traumatic incident exposure.

## METHODS
### Methodology
Qualitative evidence synthesis (QES) is a recognised method of integrating primary qualitative research findings in health and social care.[38] The methods of this QES are reported using the ENTREQ framework (online supplemental file 1). The research question and final search terms were ratified by a consensus panel of key stakeholders drawn from UK ambulance services and user advisory group of ESWs. This group contributed to the development and refinement of the review questions, search parameters and application of the review findings.

### Eligibility criteria
In keeping with qualitative review guidance,[38] the PerSPEcTiF framework was used to enhance description of inclusion criteria[39] (table 1).

### Inclusion criteria
(* indicates further information below)
(1) Study participants were front-line ESWs (studies with mixed populations of eligible participants were included); (2) The study focus was work-related psychological distress*; (3) Data collection included primary

**Table 1** The PerSPEcTiF question formulation framework[39]

| Perspective | Setting | Phenomenon of interest/problem | Environment | (Optional comparison, not applicable) | Time/timing | Findings |
|---|---|---|---|---|---|---|
| Emergency service workers | Emergency front-line ambulance, police or fire service work | Factors influencing mental well-being and help-seeking behaviour | Poor mental health outcomes and elevated rates of mental health stigma within the emergency services | | Following occupational exposure to traumatic event(s) | Emergency service workers' perceptions and experiences regarding the phenomenon of interest |

qualitative interviews, focus groups or observational methods (this included mixed-methods studies with qualitative components); (4) Analysis focused around participant attitudes towards: (A) behaviour aimed at improving or protecting mental health after experiencing a traumatic event or (B) factors which ESWs find helpful or unhelpful for their mental health while experiencing work-related psychological distress; (5) Published in English and peer-reviewed.

No limits were applied to publication date or study location.

## Exclusion criteria

1. Due to the unique nature of the traumatic events witnesses in this cohort,[40] studies investigating a military cohort were excluded.
2. Volunteer ESWs were not included.

*The term 'psychological distress will be referred to according to Ridner's definition (see online supplemental appendix A).[41]

## Search strategy

Systematic searches of the following four databases were conducted by the primary author (NA) from 1 January 1980 to March 2020: OVID MEDLINE, EMBASE, PsycINFO and SCOPUS. The first three databases were searched together using the MEDLINE database. In order to avoid duplicates, the SCOPUS database was then searched with an additional filter to exclude MEDLINE results. One reviewer (NA) screened the 13 381 article titles and/or abstracts identified by the search. The 42 full-text articles identified in this process were then independently assessed for their eligibility criteria by two reviewers (NA and RR), with the aid of RefWorks reference management software. During this process, the two reviewers agreed to exclude 17 studies which did not meet the eligibility criteria. In cases of uncertainty surrounding eligibility criteria, there was a group discussion between all authors. One additional study was excluded on the grounds of low methodological study quality; this was agreed on by all authors (figure 1).

To identify articles missed in the electronic database search, the following methods were employed: (1) Using 'related article' feature (when available), (2) Searching the titles of included studies in google scholar for citation tracking purposes and (3) Manual searching of the references of relevant studies (reference mining).

Grey literature was searched during background research for context. Two combinations of search criteria (see online supplemental appendix B) were entered into each database in order to locate relevant literature relating to help seeking and mental health recovery. The search terms were developed during a process of trial and error using qualitative guidance,[42] other reviews in the field,[37 43] and virtual consultations with stakeholders and coauthors.

## Quality appraisal

One reviewer (NA) assessed study quality using the CASP qualitative checklist.[44] Studies scoring less than five

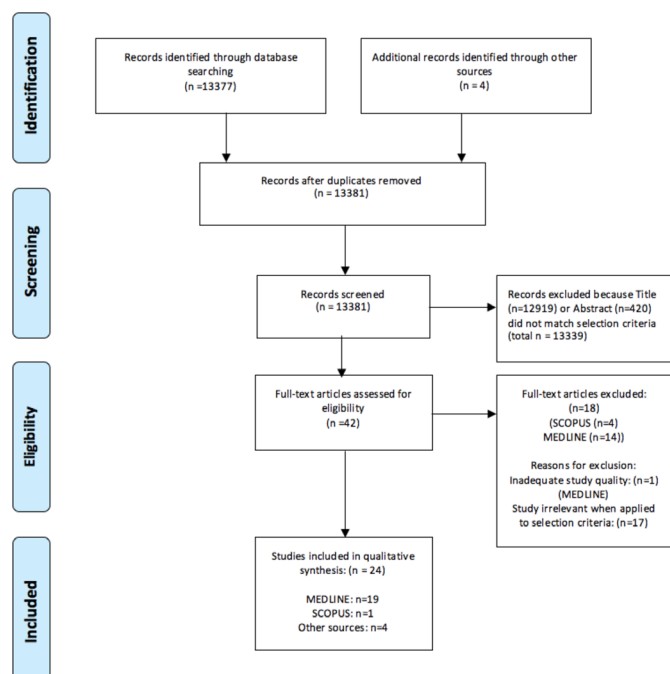

**Figure 1** Preferred Reporting Items for Systematic Reviews and Meta-Analyses flow diagram.[102]

were further appraised independently by one of three reviewers (RR, MJB and JW) to determine whether they should be included in this review based on their conceptual richness, drawing on five quality appraisal prompts as recommended by Dixon-Woods *et al.*[45] One study was removed during this approach[46] and in keeping with this approach, 'signal' (the relevance of papers to the review's aims) was prioritised over 'noise' (the inverse of methodological quality).[45]

## Data extraction

A customised data extraction template (see online supplemental appendix C) was created using qualitative data extraction guidance[42] and that of a similar review as a template.[43] In keeping with a thematic synthesis approach,[47] first order constructs (direct participant quotations) as well as the author's interpretations were extracted in separate sections to allow for a more comprehensive review.[42] Following Thomas and Harden,[47] all data relevant to the research aims were extracted into templates by one researcher (NA).

## Data synthesis

The analytical technique of thematic synthesis is an established method in the field of mental health to investigate barriers to help seeking, and to synthesise qualitative evidence about participant experiences.[37 43 48] Thematic analysis uses inductive line-by-line coding and is focused around: intervention need, appropriateness, acceptability and effectiveness.[49] It is therefore appropriate given that the purpose of the review is to inform UK ESO guidelines. The extracted text underwent line-by-line open coding, which allows new codes to emerge from the data, rather than imposing a pre-existing framework

onto the extracted data.[47] Subsequently, first-level codes were assimilated according to their meanings, similarities and differences. The codes were then arranged in a 'hierarchical tree structure' during which some codes were renamed as new meanings were generated. For example, some quotations initially coded as 'stigma' were later agreed to more appropriately reflect 'macho culture' or 'career concerns', with 'stigma and shame' relating specifically to shaming practices. This inductive process results in 'descriptive themes'. Two reviewers (NA and RR) performed the initial coding process, but grouping into descriptive themes and interpretation of the data involved discussion between all four authors. To enhance transparency, primary quotations used to construct the themes are supplied in online supplemental appendix D. In the final stage of thematic synthesis, the descriptive themes were used to address the review questions. This process involves inferring the meaning behind the data. The final result of this process was the translation of the descriptive themes into implications for ESO well-being policy practice, which involved round table discussion between all four authors.

## Patient and public involvement statement

Patients were not involved the design, conduct, dissemination or reporting of this research. However, a user advisory group of ambulance management stakeholders and ESWs was consulted regarding the design and the purpose of the research and advised on questions to include. We believed patient and public involvement to be of reduced relevance as the impetus for this review derived from the Association of Ambulance Chief Executives, who wished to address an evidence gap to inform policy decisions.

## RESULTS
### Overview of study characteristics

Twenty-four studies were included in this review (table 2). All but two studies[50 51] employed qualitative methodology only. For these two mixed-methods studies, only data from sections related to their qualitative methodology were extracted. The majority of studies (16 of 24) employed a sample of ambulance personnel,[50–65] followed by police officers[66–69] and lastly firefighters.[70 71] The participants of one study, in addition to ESWs, included participants who did not match the selection criteria,[70] and two studies included a mix of different types of ESWs.[72 73] These studies were still eligible as the participant quotations were labelled with an occupational identifier to make it possible to differentiate between eligible and non-eligible participants. The objectives of studies varied widely in terms of relevance to our review aims. Ten of the 24 studies included data relevant to protecting post-traumatic incident mental health,[51–53 55–58 61 65 73] and one study included data relevant to barriers and facilitators for help-seeking behaviour related to mental health.[72] The remaining 13 studies[50 54 59 60 62–64 66–71] included data which were extracted for both of these outcomes.

### Study quality

Study quality varied significantly with data collection, data analysis and discussion of findings being adequately explained in the majority of studies. All studies provided adequate research aims, although fewer than half sufficiently justified the research design.[51 53 55 59 60 65–67 69 71 72] The lowest scoring domains were recruitment and demonstrating reflexivity. In few studies the researchers adequately justified their selection of participants[51 55 57 67 72 73] or critically examined their relationship with participants.[58 62 63 66 66 72 73] Four studies failed to present ethical considerations.[50 51 64 70]

The full Critical Appraisal Skills Programme (CASP) table of all included studies is provided in online supplemental file 2. Following CASP appraisal, six studies[49 55 57 61 64 70] initially identified as weaker quality underwent further independent quality appraisal by one of three reviewers (RR, MJB and JW), during which process one study was excluded.[49]

This qualitative review identified 14 descriptive themes, which are arranged in groups of higher order themes. In turn, these are grouped into one of the following two overarching constructs: 'factors contributing to mental health' or 'factors influencing mental health help-seeking behaviour'. Table 3 presents a summary of hierarchical thematic structure.

### Factors contributing to mental health

The following themes describe factors which participants attribute to having a direct influence on protecting their mental health or facilitating recovery from mental ill health after traumatic incident exposure.

#### Organisational sources of support

Three themes were identified as being directly linked with systems put in place by the organisations employing ESWs and represent opportunities for mental health support following trauma exposure over which ESOs have control.

##### Time-out/downtime period

A 'time-out' or 'downtime' period refers to a period of time following a stressful call in which ESWs are temporarily placed off duty, the availability of which was inconsistent across the studies. ESWs working in organisations in which they were offered downtime by their supervisors following certain calls reported that these breaks, ranging from 30 min to 2 hours,[53 56] were essential in order to allow them to 'decompress' in preparation for the remainder of the shift.[53 54] ESWs found particular comfort in casual conversation with their colleagues during this time, which did not necessarily relate to the previous call.[53] During such discussion, humour could be employed by the group as a method of distraction and off-loading.[59] While the majority of ESWs preferred to be in the company of colleagues during this time,[55 59 60 63] some individuals chose to make use of organisation-provided exercise equipment in order to de-stress.[65]

**Table 2** Study characteristics

| Title of study (authors, year) | Quality ranking against CASP criteria | Participants | Study aim(s) | Country | Method of data collection | Analysis method |
|---|---|---|---|---|---|---|
| Halpern et al 2009[52] | 5/9 | N=60 4 supervisors, 54 front-line ambulance workers | To characterise critical incidents as well as elicit suggestions for interventions | Canada | Focus groups/ individual interviews | Ethnographic content analysis |
| Evans et al 2013[66] | 6/9 | N=19 Police officers | 1. What are police officers' experiences of supportive and unsupportive interactions following potentially traumatic incidents? 2. Do interactions differ on the basis of the context and source of support (ie, at work with colleagues and supervisors, or outside of work with family and friends)? 3. How do supportive/unsupportive interactions facilitate/hinder the processing of traumatic incidents? | England | Semistructured interviews | Thematic analysis |
| Halpern et al 2008[53] | 6/9 | N=60 4 supervisors, 54 front-line ambulance workers | To explore and describe Emergency Medical Technicians' (EMTs) experiences of critical incidents and views about potential interventions, in order to facilitate development of interventions that take into account EMS culture | Canada | Focus groups/ individual interviews | Ethnographic content analysis |
| Jones et al, 2020[72] | 9/9 | N=32 Twenty-five (78%) of the participants were active firefighters, 15 (47%) were certified EMTs, and 11 (34.4%) were certified EMTs/ paramedics. | To explore factors that influenced FRs' perceptions of mental health problems and engagement in mental health services. | USA | Ethnographic individual interviews | Content analysis |
| Regehr et al, 2002[50] | 6/9 | N=18 Paramedics | This mixed-methods study attempts to better understand factors that lead to higher levels of distress among paramedics within the theoretical framework of emotional and cognitive empathy. | Canada | Semistructured interviews | Thematic analysis |
| Jonsson and Segesten 2003[54] | 5/9 | N=362 240 EMTs, 122 registered nurses | The aim of this phenomenological study is to uncover the essence of traumatic events experienced by Swedish ambulance personnel. | Sweden | Written reports | Phenomenological analysis |
| Jonsson and Segesten 2004[55] | 4/9 | N=10 Ambulance nurses and ambulance technicians | The phenomenon approached in this study could be described as 'the way ambulance staff experience and handle traumatic events'. | Sweden | Individual interviews | Descriptive phenomenology |

Continued

**Table 2** Continued

| Title of study (authors, year) | Quality ranking against CASP criteria | Participants | Study aim(s) | Country | Method of data collection | Analysis method |
|---|---|---|---|---|---|---|
| Regehr and Millar 2007[51] | 7/9 | N=17 Paramedics | This mixed-methods study involving survey design and qualitative interviews seeks further to understand the factors related to these high levels of occupational stress. | Canada | Long interviews | Constant comparative method |
| Jeruzal et al 2019[56] | 8/9 | N=17 Paramedics and EMTs | This qualitative study was conducted to increase understanding about the difficulties of responding to paediatric calls and to obtain information about how organisations can better support EMS providers in managing potentially difficult calls. | USA | Focus groups | Directed content analysis |
| Strydom et al 2015[67] | 5/9 | N=40 Police officials | To conduct a qualitative situational analysis by exploring the experience and specific needs with regards to trauma and trauma intervention of police officials within the North-West Province's specialist units. | South Africa | Focus groups | Thematic analysis |
| Haslam and Mallon 2003[70] | 4/9 | N=31 11 firefighters, 8 station officers, 4 sub officers, 4 leading firefighters, 2 fire control officers, 2 area divisional officers | This preliminary study aimed to conduct an in-depth investigation of symptoms cited by fire service personnel and assess potential risk factors for mental health and PTSD. | England | Individual interviews | Thematic analysis |
| Fernández-Aedo et al 2017[57] | 4/9 | N=13 7EMTs, 6 ambulance nurses | To explore the experiences, emotions and coping skills among EMTs and emergency nurses after performing out-of-hospital cardiopulmonary resuscitation manoeuvres resulting in death. | Spain | Semistructured individual interviews and focus groups | Thematic analysis |
| Svensson and Fridlund 2008[58] | 7/9 | N=25 Ambulance nurses | The purpose of this study was to describe critical incidents in which ambulance nurses experience worry in their professional life and the actions they take in order to prevent and cope with it. | Sweden | Semistructured individual interviews | Structural analysis/critical incident technique |
| Clompus and Albarran 2016[59] | 5/9 | N=7 Paramedics or EMTs | The aim of this study was to explore the question of how paramedics 'survive' their work within the current healthcare climate. | England | Biographical narrative interviews and semistructured interview with all participants | Thematic analysis |
| Abelsson 2019[71] | 5/9 | N=35 Firefighters | The purpose of this paper is to describe firefighters' experiences of first response emergency care | Sweden | Group interviews | Interpretive qualitative content analysis |

Continued

**Table 2** Continued

| Title of study (authors, year) | Quality ranking against CASP criteria | Participants | Study aim(s) | Country | Method of data collection | Analysis method |
|---|---|---|---|---|---|---|
| Douglas et al [60] | 5/9 | N=28 Paramedics | To explore paramedics' experiences and coping strategies with death notification in the field. | Canada | Focus groups | Inductive thematic analysis |
| Carvello et al 2019 [61] | 4/9 | N=14 Ambulance nurses | The aim of the study is to explore the experiences, the opinions and feelings of EMS nursing staff in relation to the use of the peer supporting model. | Italy | Semistructured interviews | Not made explicit |
| Hasselqvist-Ax et al 2019 [73] | 8/9 | N=22 10 police officers, 12 firefighters | The aim of this interview study was to explore firefighters' and police officers' experiences of saving lives in out-of-hospital cardiac arrest in a dual dispatch programme. | Sweden | Individual interviews | Critical incident technique |
| Oliveira et al 2019 [62] | 8/9 | N=14 Ambulance personnel | The purpose of this paper is to explore, from this group perspective, sources of stress, coping strategies and support measures | Portugal | Semistructured interviews | Thematic analysis |
| Drewitz-Chesney 2019 [63] | 7/9 | N=8 Paramedics | The study aim was to learn about peer communication and emotional expression between paramedics in the workplace, after they respond to calls. | Canada | Semistructured individual interviews | Constructivist grounded theory |
| Edwards and Kotera 2020 [68] | 6/9 | N=5 Police officers | The study aims to explore institutional negativity and stigma in the police force towards mental ill health | UK | Semistructured individual interviews | Thematic analysis |
| Gallagher and McGilloway 2007 [64] | 2/9 | N=27 21 EMTs, 6 emergency medical controllers/ dispatchers | The principal aim of this second stage of the study was to ascertain, using qualitative methods, the impact of CIs on front-line staff by allowing them to tell their own stories. | Ireland | Individual interviews | Thematic analysis |
| Bullock and Garland 2018 [69] | 5/9 | N=59 52 police officers, 2 police staff, 4 police community support officers, one special constable | The processes through which some police officers with mental ill health experience stigmatisation. | England and Wales | Phone interviews | Thematic analysis |
| Folwell and Kauer 2018 [65] | 6/9 | N=25 EMTs | This study explores the lived experience of EMTs involved in unsuccessful paediatric resuscitation efforts and how this experience affects them professionally and personally. | USA | Individual interviews | Constant comparative analysis |

EMSs, emergency medical services; PTSD, post-traumatic stress disorder.

**Table 3** Summary of themes

| Domain | Higher level theme | Descriptive themes |
|---|---|---|
| Factors contributing to mental health | Organisational sources of support | ► Time out/downtime<br>► Supervisor<br>► Official peer support network |
| | Informal support | ► Colleagues and family<br>► Regular partner<br>► Reassurance and validation |
| Factors influencing help-seeking behaviour | Nature of intervention delivery | ► Mandatory versus non-mandatory<br>► Shared experiences with intervention provider |
| | Stigma as a barrier to help-seeking | ► Macho culture<br>► Stigma and shame<br>► Career concerns<br>► Confidentiality concerns |
| | Mental health literacy | ► Emotional awareness<br>► Education and stigma |

When time-out opportunities were unavailable, ESWs describe rushing into the next call without having psychologically processed the previous call.[51] In such circumstances, paramedics reported difficulty giving their full attention to the next call, limiting their ability to provide life-saving care.[64] Contrary to the above findings, one ambulance worker preferred to be dispatched to another call immediately following a stressful call, due to the distraction this provided.[53]

*Supervisor*
ESWs have supervisors or line managers, whose roles include acting as a point of contact after a traumatic incident. In an ambulance setting, paramedics were appreciative of the 'genuine concern' shown by their managers or supervisors following a traumatic call.[52] Concern was commonly expressed by asking paramedics how they were feeling, and providing them with an opportunity to talk.[52 53] Not all ESWs want to be approached by their supervisor immediately following the call, as illustrated by the following quote from a paramedic:

> 'I don't want you to come up and get in my face and say, are you okay? Just leave me alone. Okay. Ask me in a couple of days, am I okay with the call, sort of thing.'[53]

Occasionally supervisors were responsible for taking an ambulance crew off duty after a call.[53] Even if downtime opportunities were available on individual request, paramedics described not making use of the opportunity unless suggested or requested by the supervisor.[60] Conversely, a supervisor's influence may also dissuade paramedics from requesting temporary downtime, especially for newer paramedics who were fearful of perceived repercussions linked to an inability to cope.[60]

Paramedics described unsupportive supervisor responses, which could include failing to recognise the traumatising effect of an incident, applying disciplinary pressure after complicated calls,[51] or showing a lack of concern for paramedics' mental well-being.[53 62]

*Official peer support network*
The majority of ambulance nurses taking part in one study were in favour of peer supporters within their organisation whom the nurses described as being able to understand their distress due to their common experiences.[61] Despite the apparent popularity of the service in this organisation, peer support networks in other organisations were rarely used[59 64] with defusing occurring 'naturally within the halls' instead.[59] Participants in two studies expressed a hesitancy to make use of peer support opportunities for fear of being judged by colleagues as 'weak'.[61 64] Other concerns centred around the competence of peer supporters, their ability to maintain confidentiality,[61 64] and fear of overwhelming colleagues delivering the support.[61]

*Informal support*
In contrast to organisational factors, three themes emerged related to how informal social factors influence the protection of ESW mental health and recovery from mental ill health.

*Colleagues and family*
ESWs reported that they found it useful to talk with someone in an informal manner.[53–55 57 59–61 63 64 66 68 70 71] One firefighter described a need to 'vent a backpack', which fills up after each call.[71] Along a similar vein, one study reported suppression to be harmful in the long term since this coping style can obfuscate mental health conditions.[68] The main providers of such informal support were family members and work colleagues, but there were mixed findings in terms of preferences for support.

Many ESWs reported turning to their family members as a primary source of emotional support following difficult calls,[50 53 56 59 66 70 72] who were capable of 'selfless

listening' without judgement,[66] and with whom ESWs felt more comfortable sharing emotional vulnerability compared with colleagues.[53 66] However, a number of ESWs reported avoiding talking to their family members about stressful calls out of a wish to protect them from the trauma they experienced,[56 58 59 66 70–72] although this did not apply to family members with a first responder/healthcare background, who were judged to be able to understand ESWs' traumatic experiences.[57–59] For similar reasons, ESWs were willing to talk to certain colleagues about traumatic calls. The informal sharing of vulnerability was reserved for colleagues with whom ESWs shared a bond of trust[57 58 63 66] and for those more likely to empathise and understand the emotional impact of the event.[50 53 55–58] Sharing experiences with trusted colleagues provided an opportunity for reflection and to hear different interpretations of the event. The risks of disclosure included reliving distressing events, and potential for feelings to be invalidated when partners felt differently about the event.[58]

### Regular partner

ESWs described how having a regular work partner helped their ability to process traumatic events encountered on the job.[51 53 63 72] A trusting relationship between partners facilitated comfortable sharing of vulnerability following traumatic calls.[53 63] Having shared the experience, partners could emotionally support colleagues by allowing them to talk about the call and provide reassurance.[51 72] Having a regular partner could, however, be a negative influence in the case of an unsympathetic relationship, such as partners who respond insensitively to disclosures of vulnerability.[63] Due to the potential stigma arising from the disclosure of vulnerability within earshot of colleagues, the process of 'defusing' between partners, following a call, commonly takes place within the private space of the ambulance, when returning to base and while awaiting the next call.[63]

### Reassurance and validation

Reassurance, provided by colleagues indicating that they would have acted in the same way[52 54 58] or by receiving praise for their actions from their supervisor,[53] were valued by ESWs following traumatic incidents, especially those involving fatalities.[53] Reviewing the technical aspects of calls with other ESWs provided reassurance that the final outcome was unavoidable.[71] In cases of suicide, learning about the preceding circumstances could provide closure for some ESWs.[53] Following fatal accidents, paramedics also described needing to visit family members in hospital or to attend funerals.[52]

### Factors influencing help-seeking behaviour

The following themes reflect the barriers and facilitators to help seeking following occupational traumatic exposure.

### Nature of intervention delivery

Two themes emerged related to how the method in which a formal intervention is offered can influence attitudes towards engagement.

### Mandatory versus non-mandatory

The decision to employ optional or mandatory organisational mental health support for ESWs following traumatic calls was raised in several of the studies and often depended on the timing of delivery following the incident. Some participants resisted mandatory organisational mental health support following traumatic calls; police officers expressed a need to 'feel in control of the decision to talk' due to the stigma surrounding any disclosure of vulnerability.[66] Mandatory interventions for ESWs could lead to a rejection of the intervention,[53 60] as illustrated by the following paramedic quote:

'My emotions are none of your business and if I wanted to share my emotions with you, I'm going to share [them] with someone I trust…'[60]

Others, however, believed mandatory interventions would reduce stigma associated with their use,[60 66] and prevent delays to help seeking due to the stigma associated with disclosing vulnerability.[53] Police officers who were initially reluctant to participate felt 'calmer' and expressed gratitude after attending a mandatory counselling service.[66] EMS staff in one study suggested limiting mandatory support to certain types of incident, such as those involving children.[53]

### Shared experiences with intervention provider

Therapists with a background in the emergency services or the military, or trained peers, were preferred by ESWs[72] because of the belief that they are more likely to understand their problems and experiences.[60] This finding was observed across the emergency services:

'Many [participants] also approved of a provider that 'knew the job,' either working with multiple FRs in the past, or even as a family member.'[72]

### Stigma as a help-seeking barrier
### 'Macho' culture

Stigma associated with the disclosure of emotional vulnerability related to traumatic calls, and mental health issues,[50 53 54 59 60 62–64 66 68 70] was identified across the emergency services. A 'macho' attitude and culture acted as a key barrier to disclosure where there was an expectation to 'deal with it'.[63 68] Disclosure of vulnerability in such a culture was perceived as a weakness and responders were viewed as unable to cope with the demands of the job as a first responder.[53 59 60 63 66 69 70] Revealing one's feelings was perceived to be emasculating and prevented ESWs from talking about their feelings and seeking support[60 66 70] as demonstrated by this quote from a police officer:

I think there's a real element of machismo and masculinity in the police force and it's a bit, sort of a faux

pas to admit that things have really affected you … If I'd have come out and said 'ah you know, that really affected me badly, let's go and sit down and have a cup of tea and talk about it' I think you're straying into pink and fluffy territory there … saying that made me feel sad' is a bit too far.[66]

Stigmatising attitudes held by senior organisational members were influential as they prevented ESWs from contacting their supervisors to seek support.[66 69] Discussing stress in this culture was described as 'taboo'.[64] ESWs often avoided talking to their colleagues about their emotions following traumatic calls.[60 62] Police officers described how 'tough' colleagues working in such a culture have died by suicide.[68] Some organisations described a contrasting culture in which openness about emotional vulnerability was regarded as a strength.[71]

Two studies identified a connection between elements of the 'macho' culture described, and the gender of ESWs. Swedish ambulance nurses reported that organisations where there was a higher proportion of women fostered a culture of openness with respect to sharing vulnerability.[50] Of a small sample of seven paramedics, Clompus and AlbarranC noted that female participants were more likely than male participants to have made use of formal mental health support mechanisms.[59] The authors attribute the gender bias in accessing support in this study as being related to 'masculinised paramedic culture'.[59]

### Stigma and shame
This review identified that the fear of being shamed by being labelled as 'malingerers'[68] or 'the lazy and the lame'[69] resulted in presenteeism when officers remained on active duty although mentally unwell. Mental health stigma appears related to the belief that affected individuals are less competent in their responsibilities as an ESW as well as being unreliable.[69] Police officers described a common belief that 'you're on your own' working a shift with a colleague who has been open about the emotional impact of traumatic calls.[66] ESWs who have been open about their mental health diagnoses describe being labelled as 'mad'[67] or a 'crazy guy'.[72] Such attitudes may result in shame and the avoidance of help seeking.

### Career concerns
Four studies report that police officers delay help-seeking for mental illness due to concerns about the perceived impact of disclosure on their careers[66–69] Officers believed that being labelled with a mental health condition would obstruct career progression.[67–69] Officers feared being removed from 'public-facing operational roles', and/or feared a reduction in pay which related to being removed from front-line duities.[68] Fear of involuntary dismissal due to disclosure and help-seeking was also reported by study participants from the fire service.[70]

### Confidentiality concerns
Concerns regarding confidentiality were a barrier for ESWs to formal and informal help-seeking behaviour.

Formal support services were viewed with suspicion by police officers[67] while other officers felt they might be monitored or labelled as 'weak' if they sought a referral to well-being services.[66] Concerns about confidentiality were also raised by firefighters, who, therefore, requested preference for an anonymous counselling service outside of the brigade.[70] In the EMSs, emergency medical technicians (EMTs) expressed concerns about loss of confidentiality through the organisation-provided peer support network[64]; similar concerns were expressed by EMTs towards CISD.[65] There are also perceived risks of confidentiality breaches in informal settings as described by one paramedic:

> I think that the stigma is you have to be very careful who you tell that it bothered you or you might get judged as weak or you might get fired.[72]

Confidentiality concerns could also indirectly influence help-seeking or the provision of support. For example, ambulance supervisors wishing to put their crew on a time-out after a traumatic call could be dissuaded by knowing this information could be disclosed to dispatchers.[53]

### Mental health literacy
#### Emotional awareness
Participants and authors of the included studies recognised a need for more training and education about mental health related issues for ESWs and family members and supervisors, who may be in a better position to detect behavioural changes associated with mental ill health and could facilitate help-seeking.[52 53 64 72] Such education should focus on increasing ESWs' ability to detect emotional or behavioural changes within themselves, which could facilitate help-seeking behaviour.[52 64 64 67 72] ESWs expressed a desire to be informed about the types of emotions which could be triggered by work-related traumatic incidents,[52 64] which may reduce shame associated with help seeking.[72] Studies revealed that an inability to recognise milder mental health symptoms acted as a barrier to help seeking with participants writing them off as being 'grumpy',[68] and not recognising and admitting to emotional distress.[52] Additionally, participants were sometimes unaware of the support services available to them[63 67 72] and were unaware of the benefits of seeking help.[72]

#### Education and stigma
While stigma may indirectly change through improving general mental health awareness, authors also emphasised the value of education to reduce organisational stigma.[53] ESWs recommend such education to be delivered regularly in 'brief and efficient' classes of small groups by a peer from outside of the organisation.[72] Having an awareness of work-related mental health problems among colleagues appeared to be an important facilitator of help-seeking.[69 72] Experienced ESWs were regarded as being influential in reducing perceived stigma by giving

permission to other responders to 'open up'.[71] Police officers in one organisation, therefore, advocated for mental health 'champions'; colleagues, preferably leaders, who could model vulnerability by openly disclosing their mental illness and work-related distress.[69]

## DISCUSSION

This review synthesised 24 primary studies investigating ESW attitudes to help seeking and protection of mental health and recovery from mental ill health following trauma exposure. The synthesis generated 14 themes relating to 'factors contributing to mental health' and 'factors influencing mental health help-seeking behaviour'. Despite being grouped separately for increased clarity, these overarching constructs are interconnected. Both constructs explore the influence that senior organisational members have on the mental health of ESWs within their organisation. The influence could be positive, such as delivering educational sessions about mental ill health to reduce stigma, or negative, such as the finding that ESWs may not be given downtime after a critical incident. Another interconnecting area relates to the themes of 'macho culture' and 'stigma and shame', as these stigmatising attitudes contribute towards a hesitancy to use official peer support networks.

### Help seeking: culture/stigma

The findings of this review support quantitative findings that fears of a breach in confidentiality related to accessing help for mental ill health and associated career repercussions pose significant stigma-related barriers to help-seeking for ESWs.[37] This is important as any delays in help seeking can compound or exacerbate mental ill health.[74 75] Concerns about the impact of mental health disclosure on career progression, professional identity and competence have also been found in military personnel.[37 43 66 76 77]

Expressing emotional vulnerability, being labelled with a mental health condition and seeking help was equated with a perception of weakness, which contributed to a 'macho' culture. Similar social norms have been described in the military,[23 77] which is male dominated with high rates of mental health stigma.[37] There are, however, many other similarities between the two settings which are likely to be influential, such as 'norms and values that place a premium on self-reliance in the face of obstacles'.[37] Furthermore, these findings may reflect evidence that women are more likely to seek help about their mental health,[78] and corresponds with the literature suggesting that men in male-dominated professions are less likely than women to seek mental health support,[79] and experience higher rates of suicide.[80] While women experience a reduced risk of suicide in male-dominated professions, they experience a slightly increased risk in female-dominated professions,[80] highlighting the complexity of using occupational gender ratios to predict mental health help-seeking behaviours

and mental health outcomes. Occupations dominated by hyper-masculine stereotypes may also disadvantage men who do not identify with these values, therefore discouraging the very demographic who would challenge them.[81] Recruiting a more diverse, emotionally literate and aware workforce, with more women may challenge macho work cultures which prevent ESWs from disclosing their vulnerability.[81]

This review's finding of anticipated stigma acting as a barrier to help seeking, is supported by the findings from a systematic review which found that 'stigma can potentially lead to delayed presentation in mental healthcare' for ESWs,[37] and have an adverse influence on help seeking in civilian populations.[76] These findings are also consistent with a qualitative review of a military setting.[43]

In terms of attitudes to mental ill, our findings are consistent with Goffman who argued that stigma is defined in and enacted through socially constructed norms. The norms of what is stigmatising and what isn't are socially constructed within a large variety of contexts and have the potential to shift.[82] An effective way of challenging stigma associated with mental illness and changing negative perceptions is through a strategy of increasing contact[23 83–85] in which there is equal status, the opportunity for individuals to get to know each other, information which challenges negative stereotypes, active co-operation and pursuit of a mutual goal. Such approaches may be helpful to tackle the stigma and culture which prevent ESWs from seeking help. Elements of these factors can already be recognised in organisational strategies such as Mental Health Champions and Schwartz rounds.

### Help-seeking: education

This review identifies a demand for improving the mental health literacy of ESWs and describes the type of education ESWs believe would be appropriate. Mental health education has been shown to be effective at changing attitudes towards mental health disorders when aimed at large populations, smaller at-risk groups or at the individual level.[86 87] Antistigma education was introduced in UK emergency services as part of the 'Blue Light Programme',[7] and demonstrated that achieving antistigma change at the employee level requires sustained education efforts over a number of years.[7]

The literature suggests that certain types of events, such as paediatric fatalities, events involving multiple casualties or suicides, have a higher traumatising potential than others.[50 70 88] The evidence suggests, however, that the process of developing a post-trauma mental health disorder among ESWs is highly individual, relating to the personal history, situation and perspective of the exposed individual,[65] and ESWs can experience trauma in different ways depending on how they contextualise the victim.[50] Questionnaires taking into account individual factors or risk factors, such as the ESW's propensity to dwell,[89] are therefore a useful resource for predicting post-trauma psychopathology for exposed individuals.[90]

## Protecting mental health: social support and downtime

Consistent with the wider literature, review participants identified the importance of social support following exposure to traumatic incidents.[25 43 91 92] Downtime was positively valued by ESWs, yet is not commonly granted to ESWs in practice.[92 93] The types of mental health outcome affected by postincident downtime is disputed. Carlier *et al* identified 'insufficient time allowed by the employer to resolve the trauma' was correlated with higher PTSD scores in police officers 3 months post-trauma, but not at 12 months.[94] Having insufficient recovery time following traumatic calls has been correlated with higher emotional exhaustion[93] and psychological distress.[95] These results are in contrast with the findings of a cross-sectional quantitative survey of the psychological consequences of downtime in 217 ambulance workers, which revealed increasing periods of downtime, up to and including 1 day, to be significantly associated with lower depression scores, but not with symptoms of post-traumatic stress, burnout and stress-related physical symptoms.[96]

## Application of review findings to organisational interventions, policy and research

This review was initiated in collaboration with key ambulance management stakeholders who expressed a need for research to guide decisions about well-being interventions for front-line staff, criteria which were expanded by this review to also include fire and police organisations. Our findings have implications for three organisational interventions which are in use or available for use within ESOs.

### Trauma risk management

Consistent with qualitative findings of navy personnel regarding the implementation of TRiM,[97] this review identified a perception among ESWs that peer-support programmes are relevant and suitable for their needs, as well as concerns regarding confidentiality and competence of practitioners of peer support programmes. Peer supporters are not however regarded as possessing the same professional competence/credibility of a professional mental health professional. Of note, while review participants expressed concerns about being judged for being perceived as 'weak' by peers in a peer-support system, such a concern was not detected among naval officers towards TRiM,[97] despite widespread stigma towards help-seeking and mental health issues in the military.[43]

### Mental health champions

We identified that awareness of colleagues' mental health challenges could be an important facilitator for help seeking. The 'Blue light champions' role[98] therefore appears to be a potentially effective method for improving attitudes, although quantitative evaluation linking these roles with ESW mental health outcomes and culture change is lacking. Staff satisfaction surveys for these roles reveal a lack of support from management and insufficient time available to dedicate to the role.[99]

### Schwartz rounds

In terms of challenging organisational culture which prohibits help seeking due to stigma, Schwartz rounds offer staff the opportunity to disclose their vulnerability while fostering a connectedness to other staff. However, Schwartz Rounds have not been evaluated and it is unclear whether they reduce stigma, facilitate help-seeking behaviour, or alter mental health outcomes. Further evaluation of Schwartz Rounds is required.

## A note about COVID-19

This is the first review to qualitatively synthesise the barriers and facilitators to help seeking in the emergency services, and front-line staff's experience of what may protect and what may hinder mental health after traumatic incident exposure. The search was conducted in March 2020, before the emergence of qualitative literature relating to the COVID-19 pandemic, offering a summary of evidence preceding this significant historical benchmark. Qualitative literature relating to ESW mental health in the context of COVID-19 necessitates analysis in its own right, due to the distinct psychological stressors brought on by the pandemic, such as fear of infecting family members.[100] Healthcare workers experience different mental health pressures during pandemic working,[101] and should, therefore, be studied in a separate context to police and fire workers.

## Strengths and limitations

A core strength of this review is the analysis, which includes multiple perspectives on the topic, comprising of a primary care and ambulance clinician, a medical student and two applied health researchers. A user advisory group including ESWs was consulted regarding the design and purpose of the research and advised on questions to include, broadening its applicability. Limitations include the focus on only English language studies conducted in western countries with the exception of one study.[67] As such, findings may not generalise to other countries. Study quality varied significantly with research design commonly being inadequately explained. It should be recognised that while the review has applicability to ambulance fire and police services, the majority of studies meeting the eligibility criteria were drawn from an ambulance service background. Only one reviewer performed the initial screen of studies. However, a second reviewer was recruited to independently assess potentially relevant studies against eligibility criteria. Although the included studies are drawn from pre-Covid literature, the findings are omnirelevant to the issues of traumatisation and mental health help seeking in ESWs.

## Implications for further research to help inform policy

Following traumatic calls, ESWs will likely benefit from a recovery period during which they may wish to access informal support from their colleagues. ESOs should be aware of the therapeutic effects of informal support in the post-incident setting and facilitate its availability

since ESWs are unlikely to request downtime themselves. It may also be useful for supervisors to consider that the manner in which they approach ESWs for welfare concerns influences their help-seeking behaviours. It may be important to consider that ESWs report trusting professional relationships, such as regular work-partners, as being psychologically protective against the psychological consequences of occupational trauma experiences. ESWs may benefit from education enhancing their ability to recognise pathological emotional and behavioural changes associated with traumatic incident exposure for themselves and colleagues, and to able to suggest accessing formal organisational resources and interventions as appropriate. ESOs may wish to consider that mental health champion-type roles, providing staff in such roles are adequately supported, are regarded by ESWs as valuable tools for challenging mental health stigma. Our findings may provide insights into how engagement between ESWs and official peer support networks could be improved, by focusing on promoting antistigma and by targeting barriers to help seeking.

## CONCLUSION

Our review identified barriers and facilitators to help seeking, which may assist emergency medical organisations in improving staff engagement with organisational well-being interventions. This is in keeping with the organisation's responsibility to dismantle barriers to help seeking and reduce stigma related to mental ill health and vulnerability. Our review identified the importance of organisational cultures in which it was safe to be vulnerable and need for supportive and compassionate leadership.

**Acknowledgements** This work was initially written as a dissertation. We would like to thank Justin Waring for being a valued source of support throughout the dissertation. Special thanks to Anna Parry for her vital contributions in the early stages of the project, and involving key ambulance stakeholders during the process of developing the project aims. We would also like to thank Marcus Auth for his assistance provided in the form of advice regarding structure of the paper, and formulating the abstract. Dr Wild's research is supported by MQ, the Wellcome Trust, and the Oxford Health NIHR Biomedical Research Centre.

**Contributors** NA is the main author involved in all stages of the design, analysis and drafting and writing of the paper. RR was involved in the conception of the project's scope, design, and interpretation of results, providing important supervision and mentorship throughout. RR, MJB and JW assisted with drafting the article and revising it critically for important intellectual content, providing methodological advice, quality appraisal by independently appraising lower CASP scoring studies, and final approval of the version to be published. All authors were involved in addressing comments made during the major revision, and have approved the final version of the paper ahead of publication. As guarantor, RR accepts full responsibility for the conduct of the study, had access to the data, and controlled the decision to publish.

**Funding** The authors have not declared a specific grant for this research from any funding agency in the public, commercial or not-for-profit sectors.

**Competing interests** None declared.

**Patient consent for publication** Not applicable.

**Ethics approval** This study does not involve human participants.

**Provenance and peer review** Not commissioned; externally peer reviewed.

**Data availability statement** All data relevant to the study are included in the article or uploaded as online supplemental information. No additional data are available.

**ORCID iDs**
Niklas Maximilian Auth http://orcid.org/0000-0002-1959-6479
Jennifer Wild http://orcid.org/0000-0001-5463-1711
Ruth Riley http://orcid.org/0000-0001-8774-5344

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
