## [Reviewer comments · BMJ Open]

ARTICLE DETAILS

TITLE (PROVISIONAL)	Mental health and help-seeking among trauma-exposed emergency service staff: A qualitative evidence synthesis
AUTHORS	Auth, Niklas; Booker, Matthew; Wild, Jennifer; Riley, Ruth

VERSION 1 – REVIEW

REVIEWER	Scantlebury, Arabella University of York, York Trials Unit, Department of Health Sciences
REVIEW RETURNED	25-Mar-2021

GENERAL COMMENTS	Thank you for inviting me to review this manuscript. The topic area and findings are both important and interesting. I have focussed my review on the methodological quality of the review and unfortunately have a number of concerns, many of which are difficult to rectify at this stage. Qualitative evidence synthesis is a rapidly evolving methodology and it is a shame that the authors have not cited, or conducted their review in line with the newly published Cochrane guidance for qualitative evidence synthesis. For instance, it is recommended that reviews of qualitative evidence are referred to as qualitative evidence synthesis (Flemming and Noyes; Qualitative evidence synthesis: where are we at? 2021). The review was also not prospectively registered on PROSPERO, screening was not undertaken independently by two reviewers and there is a lack of transparency and justification surrounding the eligibility criteria and development and conduct of the search strategy. Abstract: - The review should be referred to as a qualitative evidence synthesis, in keeping with current guidance.- I appreciate abstract word counts can mean that some trade-offs need to be made. However, it would be helpful in the abstract to include further details surrounding eligibility criteria, who undertook screening, data extraction, quality appraisal.- I am not sure what is meant by 'using a thematic technique', please state the analytical method (thematic synthesis) used.- The results would benefit from greater interpretation beyond a list of themes.- Themes are described as being categorised into 'two groups'. However, describing these as overarching themes or constructs may be more appropriate. Strengths and limitations of the study: - My understanding is that these should be focussed on methodological strengths and limitations rather than summarising findings.- Bullet point 2 does not feel like a strength to me and it is unclear what is meant by correspondence. Was a formal roundtable or
--

	presentation of findings with stakeholders held or were these individuals part of the review team/ a steering group? Introduction:  - It would be helpful to readers who are unfamiliar with the topic area to describe which professional groups you are referring to when you say 'emergency service workers' up front as at the moment this comes later on in the introduction. - It would be helpful to cite references and or interventions/initiatives from outside the UK to make the paper more relevant to an international audience. Methods:  - The first paragraph is unclear to me. In particular, the description and justification for using thematic synthesis. The authors describe using an 'interpretive review technique method of thematic synthesis.' Later in the methods section the authors cite a paper by Mary Dixon Woods, which proposed 'Critical Interpretive Synthesis'. The use of terminology here is therefore making it unclear if the authors are conflating the analytical technique 'thematic synthesis' with the overarching review methodology 'qualitative evidence synthesis'. Or whether they are stating that they used the method proposed by Dixon-Woods. The latter, does not feel suited to this review and the terminology surrounding the review methods needs to be clarified. The authors also describe using an 'inductive synthesis technique', which is also confusing and again i am not sure if this is being confused with inductive coding? - There is no reference to a protocol? - In relation to the inclusion criteria, there are no exclusion criteria listed and limited details and justification of the listed inclusion criteria are provided. In keeping with current QES guidance, using the SPIDER, PERSPECTIF or SPICE frameworks to describe the eligibility criteria would be helpful. - I have particular concerns about the robustness of the search strategy and study selection process. The authors state that they searched 6 databases, but four are listed. I am not sure what is meant by the 'first three databases were searched together'. Does this mean that the same search terms were applied to each database as not using the MESH headings that are relevant for each database would greatly affect the number of records you identify? Or does this mean that the databases were all searched using OVID as this is also problematic? - It is also unclear who developed the search strategy (information specialist, member of the review team) and how this was developed (based on previous search strategies of related literature). I am unclear as to who conducted screening of title, and abstract and full text. It is a shame that this was only conducted by one reviewer. - Searches were conducted in March 2020 and so may need updating. - What reference management software was used? - No details of any year or country restrictions that were applied to the searches are provided. - The authors have chosen to exclude studies based on quality and cite Mary Dixon-Woods' paper. However, this paper was published some time ago, before the development of the new Cochrane guidance for qualitative evidence synthesis. The paper proposes a review methodology 'interpretive reviews', which focusses on critiquing the literature and generating theories or concepts. This does not seem to tally with the present review
--	--

	which is largely descriptive in nature and so more suited to a thematic synthesis.  - Although a data extraction template is provided as an appendix, there is no details of how this was developed or piloted. Including some details of the information to be extracted in the main body of the text would be useful. - I am not sure what is meant by one reviewer independently assessed study quality – independent to what? I am unsure why the authors have chosen to exclude studies based on quality. The authors state that papers were given scores relating to their study quality, with those less than five subjected to further appraisal – how were scores calculated and why was a score of five chosen as a cut off? - No details of who undertook data extraction and whether this was undertaken independently by multiple researchers are provided. - PRISMA flow diagram would be better placed at the start of the results section. - There are some inaccuracies in the flow diagram – please check the numbers add up correctly. It would also be helpful to add more specific reasons for exclusion such as number excluded based on population, methods. The diagram should also be broken down further into records title screened, records abstract screened and full text screened. - The description of the synthesis is quite ‘text booky’. How were themes translated into implications for ESO wellbeing policy, did this involve roundtable discussion/consultation with experts etc. Also, giving some examples of for example codes that were renamed or how themes evolved would help to give a sense of the process. Was coding inductive and/or deductive? - Who undertook analysis and what is their background (reflexivity). Was any data management software used to facilitate analysis (NVIVO). Results  - This is the first mention that studies with mixed populations, or qualitative elements of mixed methods studies were included. This should be discussed in the eligibility criteria - Table 1: it is difficult to understand the quality appraisal score without a denominator. As I am not familiar with the area the term supervisor is not clear to me. - Theme names are very superficial and need some further clarity. For instance, describing a theme as ‘organisational’ does not really give a sense of what is to come or help to guide the reader. - The sign posting relating to each theme and ‘domain’ is also quite superficial. It would add greater depth to the synthesis to provide some more high level interpretation at the start of each section and also help the narrative to describe any potential overlap between the factors to both mental health recovery and help-seeking as I would expect there to be some. At the moment themes are presented in isolation and whilst I acknowledge that the nature of the research question means that the findings are likely to be descriptive, a qualitative evidence synthesis usually enables a higher level of interpretation. Discussion:  - The discussion is interesting, and makes some bold statements and suggestions for practice and policy. However, the length of the discussion seems to dilute these somewhat. - I really liked the idea of applying the review findings to current organisational interventions, policy and research. However, this section, is quite hard to follow for people who are unfamiliar with
--	--

	the area and assumes a level of knowledge surrounding some of these interventions that I do not have.  - There is no strengths and limitations section. - Given the length of the discussion the conclusion would benefit from some expansion and higher level summing up of the review
--	--

REVIEWER	Case, Rosalind Monsh University, Department of Epidemiology and Preventive Medicine
REVIEW RETURNED	07-Apr-2021

GENERAL COMMENTS	Comments to Author Ms. Ref. No.: bmjopen-2020-047814 Title: The barriers and facilitators to help-seeking and recovery among psychologically distressed emergency service staff: A thematic synthesis. Overview and General Recommendation This manuscript reports on a review of qualitative studies exploring emergency service workers' (ESW) views regarding mental health support barriers and facilitators following workplace trauma exposure. The language used in the title and throughout needs refining in order to clarify the focus, however, as is further discussed below. This is an interesting piece of work that is worthy of publication; however, minor revisions are recommended in order to clarify the aims and ensure the scope and implications are realistically presented. TITLE The word 'recovery' is something of a misnomer here and at times within the body of the document, as this term is primarily used within the psychological field to refer to the recovery from a mental illness; that is, the process of rehabilitation following a period of mental ill health. It implies that one has 'lost' their mental health (i.e. in order to be able to 'recover' it). The majority of people exposed to trauma do not develop a mental disorder and thus have nothing to recover from; further, recovery refers to a longer-term process than it seems is meant here. Further consideration of a more appropriate term is recommended. It would seem that the aim of the short-term, acutely-focused interventions noted here is rather preventative (or protective),
---

	focused on de-escalation, restoration, reduction of physiological arousal and self-care. Further, the term 'psychologically distressed staff' might also be inaccurate – do we know that the participants in these studies were all distressed? Trauma-exposed? Being exposed to distressing incidents does not mean that a worker is psychologically distressed. ABSTRACT Study aim is unclear in abstract. Greater clarity and detail is warranted to highlight rationale for qualitative synthesis. While the background notes that “there is uncertainty around which organisational interventions should be provided”, the study methodology is not able address this problem as it does not enable a review of the efficacy of such interventions. It seems to go part of the way towards understanding perceptions of acceptability and usefulness, however, which are critical aspects of organisational intervention effectiveness. Delineating what is under review here seems important. The design and methodology sections would benefit from more information. It is unclear what type of thematic synthesis is employed and more procedural details would be helpful. ARTICLE SUMMARY Limitations are not adequately addressed here. BACKGROUND The introduction section is concise and mostly well written. There are some issues with structure, however, and some relatively minor semantic/style issues to note: P5 L29 – The sentence including “... the second greatest cause for poor mental health” is problematic. If left intact, the word 'for' should be substituted for 'of'. However, of arguably greater significance is the use of the word 'cause'. I am not aware of any RCTs that have evaluated the impact of trauma exposure nor compared its impact to that of workload. The reference cited does not seem to be of specific relevance to the assertion. There is a greater body of evidence that could be referred to here regarding the correlates of poor mental health that could be referred to when talking about risk factors or associations, but I am not aware of any experimental data that clearly demonstrates a causal relationship. P5 L31 – Sentence regarding definition of critical incident seems out of place and doesn't link to previous of following sentences.
--	---

From here to the end of this paragraph (L40), the structure devolves and seems bitsy and a bit random. The author seems unsure about what points they are emphasising in the introduction and it jumps around a lot.

P5 L42 – An introductory sentence would help provide context for the introduction of this brief overview of a few select intervention / preventative approaches. How/why were these approaches selected for focus here?

P6 L9 – It might be a bit ambitious to state that the findings will assist guideline development. They might aim to contribute to that objective, but can't be unilaterally stated as doing so. Given best practice for clinical guideline development relies heavily on the systematic review of quantitative studies of intervention efficacy, it seems important not to overstate the role of a narrative review of qualitative data relating to stakeholder perceptions in that process.

METHODS

Methodology section succinctly and thoroughly describes the key aspects of the methods employed.

Again, the purpose of the study to inform guidelines might be more cautiously described (here and throughout).

RESULTS

The overview of study characteristics and study quality sections are clear and easy to follow. Themes impress as intuitive and clearly derived from qualitative data synthesis. Writing in this section is concise and well-organised.

DISCUSSION

The relevance of gender constructs is pertinent and discussion of this could be further developed with reference to literature cited in our recent rapid review - Case et al. 2020 (https://www.saxinstitute.org.au/wp-content/uploads/20.10_Evidence-Check_Suicide-prevention-in-high-risk-occupations.pdf[RC1]). It seems that the qualitative data and themes of discussion identified by ESW are in many instances reflected in the quantitative evidence regarding mental health and suicide risk among these professions.

While perhaps not directly emerging from your review's findings, highlighting the risks associated with CISD is probably warranted given its mention. Many organisations are unaware of the potential harm associated with group CISD and the evidence indicates we should desist from this practice (in its traditional group format).

	While only briefly noted, it is of interest that ESW have not highlighted it as useful. Discussing findings as 'implications for policy' is perhaps an overstretch. Interesting, and potentially able to inform further research, but there is not enough high quality evidence or specific detail here to inform policy and guidelines. Highlighting the findings as reflecting the perceptions of ESW's is important; this might help to narrow the focus and aim. It might also serve to emphasise the importance of stakeholder perceptions and attitudes when considering broader utility and effectiveness of interventions. [RC1]
--	--

VERSION 1 – AUTHOR RESPONSE

Reviewer 1: Dr. Arabella Scantlebury, University of York

Thank you for inviting me to review this manuscript. The topic area and findings are both important and interesting. I have focussed my review on the methodological quality of the review and unfortunately have a number of concerns, many of which are difficult to rectify at this stage.

Thank you Dr Scantlebury for recognising the importance of this work.

Qualitative evidence synthesis is a rapidly evolving methodology and it is a shame that the authors have not cited, or conducted their review in line with the newly published Cochrane guidance for qualitative evidence synthesis. For instance, it is recommended that reviews of qualitative evidence are referred to as qualitative evidence synthesis (Flemming and Noyes; Qualitative evidence synthesis: where are we at? 2021).

Thank you for bringing this relevant guidance to our attention. This manuscript was submitted in December 2020 and therefore this guidance was not available to us at the time of writing. This guidance has been used in conjunction with your comments as a basis for changes to the manuscript.

The review was also not prospectively registered on PROSPERO,

We appreciate that the recent QES guidance referenced states 'For a QES that has a health related focus/outcome, it can be registered on the PROSPERO international database of prospectively registered systematic reviews'. This work was initially intended as an undergraduate dissertation for the University of Birmingham, a protocol was submitted to the University in December 2019 for assessment. However, on undertaking the review, it became clear that this review addressed an important gap in the literature, and that its publication would meaningfully advance the discourse on the topic.

screening was not undertaken independently by two reviewers and there is a lack of transparency and justification surrounding the eligibility criteria and development and conduct of the search strategy.

We apologise for omitting to include that a second reviewer independently reviewed the 42 potentially relevant articles identified by the search. This has now been clarified in the 'search strategy' section.

In keeping with QES guidance referred to by reviewer 1, the PERsPEcTiF question formulation framework has now been applied to the eligibility criteria to facilitate the understanding the eligibility criteria and study aims. Additionally, exclusion criteria have been added, and additional justification around eligibility criteria have been applied.

A sentence has been added for transparency on how the search criteria were created.

Abstract:

- The review should be referred to as a qualitative evidence synthesis, in keeping with current guidance.

Thematic synthesis has been changed to qualitative evidence synthesis in the title and abstract.

I appreciate abstract word counts can mean that some trade-offs need to be made. However, it would be helpful in the abstract to include further details surrounding eligibility criteria, who undertook screening, data extraction, quality appraisal.

These details have been added.

I am not sure what is meant by 'using a thematic technique', please state the analytical method (thematic synthesis) used.

We apologise for the confusing wording. 'Thematic technique' has been changed to 'thematic synthesis.'

The results would benefit from greater interpretation beyond a list of themes.

We thank the reviewer for this comment, and have reworded this section as well as adding some further interpretation in the conclusion of the abstract. We would argue that the majority of the interpretation of the results section takes place in the discussion, such as when applying the results to organisational interventions. We decided to display the results as a list of themes, in order to facilitate the process of applying the results to organisational interventions in the discussion, which is where we would argue the majority of the interpretation of the results takes place. A section highlighting some of the interconnecting features between themes has been added at the beginning of the discussion, further adding to the interpretation of the results.

Themes are described as being categorised into 'two groups'. However, describing these as overarching themes or constructs may be more appropriate.

Thematic categories have now been described as overarching constructs in the abstract and discussion. A section highlighting some of the interconnecting features between themes has been added at the beginning of the discussion.

Strengths and limitations of the study:

- My understanding is that these should be focussed on methodological strengths and limitations rather than summarising findings.

Bullet point 2 does not feel like a strength to me and it is unclear what is meant by correspondence. Was a formal roundtable or presentation of findings with stakeholders held or were these individuals part of the review team/ a steering group?

The section has been rewritten to focus on methodological strengths and weaknesses, in keeping with BMJ Open guidelines for authors. Bullet point 2 has now been removed, but there was a formal

roundtable discussion with stakeholders during the early stages of the project which helped to sculpt the project's aims.

Introduction:

- It would be helpful to readers who are unfamiliar with the topic area to describe which professional groups you are referring to when you say 'emergency service workers' up front as at the moment this comes later on in the introduction.

The first sentence in the introduction has been amended.

- It would be helpful to cite references and or interventions/initiatives from outside the UK to make the paper more relevant to an international audience.

A short section has been added to provide additional context before introducing UK interventions. This includes a review of international interventions (Antony et al. Systematic Reviews (2020) 9:121), as well as numerous references related to critical incident stress debriefing (CISD) which has widespread international use.

Methods:

- The first paragraph is unclear to me. In particular, the description and justification for using thematic synthesis. The authors describe using an 'interpretive review technique method of thematic synthesis.' Later in the methods section the authors cite a paper by Mary Dixon Woods, which proposed 'Critical Interpretive Synthesis'. The use of terminology here is therefore making it unclear if the authors are conflating the analytical technique 'thematic synthesis' with the overarching review methodology 'qualitative evidence synthesis'. Or whether they are stating that they used the method proposed by Dixon-Woods. The latter, does not feel suited to this review and the terminology surrounding the review methods needs to be clarified. The authors also describe using an 'inductive synthesis technique', which is also confusing and again I am not sure if this is being confused with inductive coding?

Thank you for highlighting the potential confusion with our description of the analytic and coding process. The analytical technique of thematic synthesis was used to conduct this review. We cite the Mary Dixon Woods paper in the quality appraisal section as a reference only to the exclusion prompts mentioned in this paper. The term 'inductive synthesis technique' has been reworded; it was originally referring to the inductive line-by-line coding which is described by Thomas and Harden as part of the thematic synthesis approach, but recognise the potential confusion that our use of these terms might have resulted in.

- There is no reference to a protocol?

Please refer to the earlier response to the comment regarding PROSPERO.

In relation to the inclusion criteria, there are no exclusion criteria listed and limited details and justification of the listed inclusion criteria are provided. In keeping with current QES guidance, using the SPIDER, PERSPECTIF or SPICE frameworks to describe the eligibility criteria would be helpful.

The PERSPECTIF question formulation framework has been applied to the eligibility criteria to provide more detail. Exclusion criteria have been added, and additional justification around eligibility criteria have been applied. For any uncertainty around eligibility criteria, there was a group discussion between all authors.

I have particular concerns about the robustness of the search strategy and study selection process. The authors state that they searched 6 databases, but four are listed. I am not sure what is meant by the 'first three databases were searched together'. Does this mean that the same search terms were applied to each database as not using the MESH headings that are relevant for each database would greatly affect the number of records you identify? Or does this mean that the databases were all searched using OVID as this is also problematic?

We apologise for this confusion; only four databases were searched. OVID MEDLINE, EMBASE and PsycINFO were searched together using OVID. Scopus was searched separately using the MESH headings for its own database.

It is also unclear who developed the search strategy (information specialist, member of the review team) and how this was developed (based on previous search strategies of related literature). I am unclear as to who conducted screening of title, and abstract and full text. It is a shame that this was only conducted by one reviewer.

A sentence has been added providing more transparency for how the search strategy was developed. We omitted to include that that a second reviewer (Dr Ruth Riley) independently reviewed the 42 full text articles, identified by NA, against their eligibility criteria. During this process, the two reviewers (Niklas Auth and Dr Ruth Riley) agreed to exclude 17 studies which did not meet the eligibility criteria. One additional study was excluded on grounds of insufficient study quality; this was agreed upon by all authors.

Searches were conducted in March 2020 and so may need updating.

We thank the reviewer for raising this important point. At the time of initial submission, an update search revealed no new studies that were not directly related to Covid pandemic. We agreed that the timing of the initial search is significant in the context of the Covid pandemic. We have therefore added a short paragraph at the end of the discussion which addresses this.

What reference management software was used?

This has been added in the 'search strategy' section of the methods.

No details of any year or country restrictions that were applied to the searches are provided.

The eligibility criteria section has been rewritten to include restrictions of year and country.

The authors have chosen to exclude studies based on quality and cite Mary Dixon-Woods' paper. However, this paper was published some time ago, before the development of the new Cochrane guidance for qualitative evidence synthesis. The paper proposes a review methodology 'interpretive reviews', which focusses on critiquing the literature and generating theories or concepts. This does not seem to tally with the present review which is largely descriptive in nature and so more suited to a thematic synthesis. I am not sure what is meant by one reviewer independently assessed study quality – independent to what? I am unsure why the authors have chosen to exclude studies based on quality. The authors state that papers were given scores relating to their study quality, with those less

than five subjected to further appraisal – how were scores calculated and why was a score of five chosen as a cut off?

The review and analysis was undertaken before the publication of the new Cochrane guidance for QES. A modified version of the CASP appraisal tool was used to assess studies for methodological quality. Studies scoring 5 or below were circulated among the review team, who decided to include all but one of these studies on the basis of the conceptual richness they offer, drawing upon the prompts by MDW to identify features consistent with a fatally flawed study. We thank the reviewer for highlighting that the word independent was used confusingly here, and has now been removed.

A focussed qualitative sensitivity analysis performed in response to this comment indicated that the paper which was excluded on the grounds of quality would not have altered the thematic structure in any meaningful way.

Although a data extraction template is provided as an appendix, there is no details of how this was developed or piloted. Including some details of the information to be extracted in the main body of the text would be useful.

We have now included details about how the extraction template was developed.

No details of who undertook data extraction and whether this was undertaken independently by multiple researchers are provided.

Sentences have been added in the data extraction section to clarify that data extraction was undertaken by one researcher, and to enhance transparency of how the synthesis process was carried out.

PRISMA flow diagram would be better placed at the start of the results section. There are some inaccuracies in the flow diagram – please check the numbers add up correctly. It would also be helpful to add more specific reasons for exclusion such as number excluded based on population, methods. The diagram should also be broken down further into records title screened, records abstract screened and full text screened.

We thank the reviewer for highlighting these inaccuracies. Figure 1 has been moved to the start of the results section, and the numbers have been amended. Unfortunately, whilst focussing on the key reportable numbers in the PRISMA chart, we have not retained the data that would enable reporting of the categories of exclusion as suggested, but do accept the value of this suggestion.

The description of the synthesis is quite 'text booky'. How were themes translated into implications for ESO wellbeing policy, did this involve roundtable discussion/consultation with experts etc. Also, giving some examples of for example codes that were renamed or how themes evolved would help to give a sense of the process. Was coding inductive and/or deductive?

We have made some amendments to the data synthesis section; we have now specified that this was an inductive process, and that roundtable discussion between all four authors was involved when developing implications for ESO wellbeing policy. We have also given examples of how codes were renamed to result in the final descriptive themes.

Who undertook analysis and what is their background (reflexivity). Was any data management software used to facilitate analysis (NVIVO).

Further information has been added to the methods section to clarify who undertook analysis. With regards to reflexivity, a sentence has been added in the strengths and limitations section. No data management software was used to facilitate analysis.

Results

- This is the first mention that studies with mixed populations, or qualitative elements of mixed methods studies were included. This should be discussed in the eligibility criteria

These changes have been made to the eligibility criteria.

Table 1: it is difficult to understand the quality appraisal score without a denominator.

A denominator has been added to the quality appraisal score in table 1.

As I am not familiar with the area the term supervisor is not clear to me.

A basic description of the term supervisor has been added to the results section: 'ESWs have supervisors/line managers, whose roles include acting as a point of contact after a traumatic incident.' A description of the term supervisor has also been added to Appendix D: Abbreviations and definitions.

Theme names are very superficial and need some further clarity. For instance, describing a theme as 'organisational' does not really give a sense of what is to come or help to guide the reader.

We thank the reviewer for highlighting this, and fully acknowledge the ethos behind the comment that overly reductive theme names can miss some of the richness and nuance of the analysis. After revisiting the coding framework, and further detailed discussion amongst the author team on this point, we have opted to retain the names but add some further descriptive clarity to the associated text where appropriate to try and help signpost the reader to the positioning of these themes within the architecture of our analysis.

The sign posting relating to each theme and 'domain' is also quite superficial. It would add greater depth to the synthesis to provide some more high level interpretation at the start of each section and also help the narrative to describe any potential overlap between the factors to both mental health recovery and help-seeking as I would expect there to be some. At the moment themes are presented in isolation and whilst I acknowledge that the nature of the research question means that the findings are likely to be descriptive, a qualitative evidence synthesis usually enables a higher level of interpretation.

We thank the reviewer for these comments. We would argue that the results do undertake a degree of interpretation, such as the discussion of gender constructs within the theme 'macho' culture. As mentioned in response to an earlier comment, we decided to display the results as a list of themes, in order to facilitate the process of applying the results to organisational interventions in the discussion, which is where we would argue the majority of the interpretation of the results takes place. A section highlighting some of the interconnecting features between themes has been added at the beginning of the discussion, further adding to the interpretation of the results.

Discussion:

- The discussion is interesting, and makes some bold statements and suggestions for practice and policy. However, the length of the discussion seems to dilute these somewhat.

A similar comment was made by reviewer 1, and we have now rephrased the title of this section to highlight that the qualitative findings themselves open avenues for areas of research, which in turn may inform policy and guidelines.

I really liked the idea of applying the review findings to current organisational interventions, policy and research. However, this section, is quite hard to follow for people who are unfamiliar with the area and assumes a level of knowledge surrounding some of these interventions that I do not have.

We are grateful for this feedback. The recommendations were based on multiple stakeholder input, which included wellbeing representatives from the ambulance service, as well as co-authors with professional academic interests in this field. We have reviewed the glossary (Appendix D) to make sure we have defined key terms for the general reader within this section; a definition for 'mental health champions' has now been added.

There is no strengths and limitations section.

A strengths and limitations section has now been added

Given the length of the discussion the conclusion would benefit from some expansion and higher level summing up of the review

A sentence has been added at the beginning of the conclusion to provide some further summing up of the review, and another sentence has been added for expansion.

Reviewer 2: Dr. Rosalind Case, Monsh University

Overview and General Recommendation

This manuscript reports on a review of qualitative studies exploring emergency service workers' (ESW) views regarding mental health support barriers and facilitators following workplace trauma exposure. The language used in the title and throughout needs refining in order to clarify the focus, however, as is further discussed below. This is an interesting piece of work that is worthy of publication; however, minor revisions are recommended in order to clarify the aims and ensure the scope and implications are realistically presented.

We are appreciative and grateful for this feedback, Dr Case.

TITLE

The word 'recovery' is something of a misnomer here and at times within the body of the document, as this term is primarily used within the psychological field to refer to the recovery from a mental illness; that is, the process of rehabilitation following a period of mental ill health. It implies that one has 'lost' their mental health (i.e. in order to be able to 'recover' it). The majority of people exposed to trauma do not develop a mental disorder and thus have nothing to recover from; further, recovery refers to a longer-term process than it seems is meant here. Further consideration of a more appropriate term is recommended. It would seem that the aim of the short-term, acutely-focused

interventions noted here is rather preventative (or protective), focused on de-escalation, restoration, reduction of physiological arousal and self-care. Further, the term 'psychologically distressed staff' might also be inaccurate – do we know that the participants in these studies were all distressed? Trauma-exposed? Being exposed to distressing incidents does not mean that a worker is psychologically distressed.

We thank the reviewer for these comments. We agree that the term 'protection' is often more appropriate than the word 'recovery' during this review, particularly in the context directly following trauma-exposure. We still believe that at times 'recovery' is appropriate, such as in the case of ESWs describing PTSD symptoms as a result of occupational trauma-exposure. Therefore, at times the word recovery is used alongside the word protection in this manuscript, where the use both words is necessary.

We also thank the reviewer for raising the important point regarding the terminology 'psychologically distressed'. We accept that trauma-exposure does not equate to psychological distress, and accept that this was not explicitly highlighted in the manuscript; this point has now been emphasised in the introduction. The term 'trauma-exposed' has been added throughout the manuscript where appropriate, occasionally replacing the term distressed where this is appropriate. With similarity to the term 'recovery', we would argue that it is important to include the insights from those ESWs who are distressed, given the disproportionately higher rate of psychological distress in these occupations. We therefore selected for studies with a 'focus' on psychological distress, yet we still included the insights of staff who were trauma-exposed and not distressed.

ABSTRACT

Study aim is unclear in abstract. Greater clarity and detail is warranted to highlight rationale for qualitative synthesis. While the background notes that "there is uncertainty around which organisational interventions should be provided", the study methodology is not able address this problem as it does not enable a review of the efficacy of such interventions. It seems to go part of the way towards understanding perceptions of acceptability and usefulness, however, which are critical aspects of organisational intervention effectiveness. Delineating what is under review here seems important. The design and methodology sections would benefit from more information. It is unclear what type of thematic synthesis is employed and more procedural details would be helpful.

A sentence has been added in the conclusion of the abstract demonstrating the rationale for a qualitative synthesis. The background of the abstract has been reworded to highlight that there is an inconsistency in the type of organisational interventions offered to trauma-exposed ESWs, without implying that this review aims to inform which interventions should be used. More procedural details about the synthesis approach and methodology in general have been added.

ARTICLE SUMMARY Limitations are not adequately addressed here.

The strengths and limitations section has now been rewritten.

1) P5 L29 – The sentence including "... the second greatest cause for poor mental health" is problematic. If left intact, the word 'for' should be substituted for 'of'. However, of arguably greater significance is the use of the word 'cause'. I am not aware of any RCTs that have evaluated the impact of trauma exposure nor compared its impact to that of workload. The reference cited does not seem to be of specific relevance to the assertion. There is a greater body of evidence that could be referred to here regarding the correlates of poor mental health that could be referred to when talking about risk factors or associations, but I am not aware of any experimental data that clearly demonstrates a causal relationship.

We apologise for the inaccurate wording of this sentence. This sentence has now been rewritten to reflect that the reference cited is a survey in which ESWs reported traumatic event exposure as a cause of poor mental health, but does not prove a causal relationship. The sentence is referring to page 12 of the pdf document (reference 6).

2) P5 L31 – Sentence regarding definition of critical incident seems out of place and doesn't link to previous or following sentences. From here to the end of this paragraph (L40), the structure devolves and seems bitsy and a bit random. The author seems unsure about what points they are emphasising in the introduction and it jumps around a lot.

The sentence structure in this paragraph has been amended into a more logical order. We have emphasised the point that PTSD (one example of the types of mental health disorders which ESWs are disproportionately affected by) is associated with an increased risk of suicidal ideation. To strengthen this point, we have additionally referenced a report recommended by Reviewer 2, which reports on a study about firefighters: 'Post-traumatic stress symptoms were more strongly predictive of suicidal ideation, plans or attempts than depression or sociodemographic variables.'

3) P5 L42 – An introductory sentence would help provide context for the introduction of this brief overview of a few select intervention / preventative approaches. How/why were these approaches selected for focus here?

Three introductory sentences have been added to provide more context, including an explanation for how these approaches were selected for focus which precedes the introduction of TRiM and Schwartz rounds.

P6 L9 – It might be a bit ambitious to state that the findings will assist guideline development. They might aim to contribute to that objective, but can't be unilaterally stated as doing so. Given best practice for clinical guideline development relies heavily on the systematic review of quantitative studies of intervention efficacy, it seems important not to overstate the role of a narrative review of qualitative data relating to stakeholder perceptions in that process.

We thank the reviewer for this comment, this sentence has been reworded.

DISCUSSION The relevance of gender constructs is pertinent and discussion of this could be further developed with reference to literature cited in our recent rapid review - Case et al. 2020 (https://www.saxinstitute.org.au/wp-content/uploads/20.10_EvidenceCheck_Suicide-prevention-in-high-risk-occupations.pdf).

We thank the reviewer for recognising the importance of gender constructs in this topic, and we have expanded upon this phenomenon within the paragraph 'Help-seeking: culture/stigma', in part using references cited by the recommended report.

It seems that the qualitative data and themes of discussion identified by ESW are in many instances reflected in the quantitative evidence regarding mental health and suicide risk among these professions. While perhaps not directly emerging from your review's findings, highlighting the risks

associated with CISD is probably warranted given its mention. Many organisations are unaware of the potential harm associated with group CISD and the evidence indicates we should desist from this practice (in its traditional group format). While only briefly noted, it is of interest that ESW have not highlighted it as useful.

We thank the reviewer for these comments. We have now added a sentence outlining the risks of CISD, although we believe that this reads more logically in the introduction, and has therefore been inserted into the paragraph describing existing organisational interventions. This is also relevant to reviewer 2's earlier comment regarding P5 L42, as there is now more context to explain why certain interventions have been selected for focus.

Discussing findings as 'implications for policy' is perhaps an overstretch. Interesting, and potentially able to inform further research, but there is not enough high quality evidence or specific detail here to inform policy and guidelines. Highlighting the findings as reflecting the perceptions of ESW's is important; this might help to narrow the focus and aim. It might also serve to emphasise the importance of stakeholder perceptions and attitudes when considering broader utility and effectiveness of interventions.

We thank the reviewer for these comments. We have rephrased the title of this section to highlight that the qualitative findings themselves open avenues for areas of research, which in turn may inform policy and guidelines. We agree that the section previously titled 'implications for policy' remains firmly grounded in the findings, and it has now been emphasised in the introduction that the findings of this review are grounded in the perceptions of ESWs.

New references: to be added at end

New references

Antony J, Brar R, Khan P, Ghassemi M, Nincic V, Sharpe J et al. Interventions for the prevention and management of occupational stress injury in first responders: a rapid overview of reviews. *Systematic Reviews*. 2020;9(1).

Australian Bureau of Statistics. National Survey of Mental Health and Wellbeing: Summary of Results, Australia, 2007. ABS cat. no. 4326.0. Canberra: ABS; 2008.

Milner A, King T. Men's work, women's work and suicide: a retrospective mortality study in Australia. *Australian and New Zealand Journal of Public Health*. 2018;43(1):27-32.

Milner A, Scovelle A, King T. Treatment-seeking differences for mental health problems in male- and non-male-dominated occupations: evidence from the HILDA cohort. *Epidemiology and Psychiatric Sciences*. 2018;28(6):630-637.

Flemming K, Noyes J. Qualitative Evidence Synthesis: Where Are We at?. *International Journal of Qualitative Methods*. 2021;20:160940692199327.

Booth A, Noyes J, Flemming K, Moore G, Tunçalp Ö, Shakibazadeh E. Formulating questions to explore complex interventions within qualitative evidence synthesis. *BMJ Global Health*. 2019;4(Suppl 1):e001107.

Litz BT. Distinct Trauma Types in Military Service Members Seeking Treatment for Posttraumatic Stress Disorder. *J Trauma Stress* 2018;31(2):286-296.

Brown University- Roles And Responsibilities Of Supervisory Staff. [internet] 2021[cited 2021 June 1]. Available from: <https://www.brown.edu/campus-life/health/ems/roles-and-responsibilities-supervisory-staff#:~:text=EMS%20Supervisors&text=The%20EMS%20Supervisor%20is%20the,and%20appropriate%20emergency%20vehicle%20operation>.

Case R, Alabakis J, Bowles K-A, Smith K: High-risk occupations—Suicide: an Evidence Check rapid review brokered by the Sax Institute (www.saxinstitute.org.au) for the NSW Ministry of Health, 2020.

Cai H, Tu B, Ma J, Chen L, Fu L, Jiang Y et al. Psychological impacts and coping strategies of front-line medical staff during COVID-19 outbreak in Hunan, China. *Medical Science Monitor*. 2020;26.

Vagni M, Maiorano T, Giostra V, Pajardi D. Coping With COVID-19: Emergency Stress, Secondary Trauma and Self-Efficacy in Healthcare and Emergency Workers in Italy. *Frontiers in Psychology*. 2020;11.

Smith A, Roberts K. Interventions for post-traumatic stress disorder and psychological distress in emergency ambulance personnel: a review of the literature. *Emergency Medicine Journal* 2003 /01/01;20(1):75-78.

Carlier IVE, Lamberts RD, Van Uchelen AJ, Gersons BPR. Disaster-related post-traumatic stress in police officers: a field study of the impact of debriefing. *Stress Med* 1998 9807 9807;14(3):143-148.

Sattler DN, Boyd B, Kirsch J. Trauma-exposed Firefighters: Relationships among Posttraumatic Growth, Posttraumatic Stress, Resource Availability, Coping and Critical Incident Stress Debriefing Experience. *Stress Health* 2014;30(5):356-365.

Tuckey MR. Group critical incident stress debriefing with emergency services personnel: a randomized controlled trial. *Anxiety, Stress & Coping* 2014;27(1):38-55.

Regel S. Post-trauma support in the workplace: the current status and practice of critical incident stress management (CISM) and psychological debriefing (PD) within organizations in the UK. *Occupational Medicine* 2007 709 0;57(6):411-416.

Rose S, Bisson J, Churchill R, Wessely S. Psychological debriefing for preventing post traumatic stress disorder (PTSD). *Cochrane Database of Systematic Reviews*. 2002;.

VERSION 2 – REVIEW

REVIEWER	Scantlebury, Arabella University of York, York Trials Unit, Department of Health Sciences
REVIEW RETURNED	01-Jul-2021

GENERAL COMMENTS	Thank you for submitting your revised version of this manuscript and for considering my revisions. Unfortunately, I still have some concerns surrounding the robustness of this review. In particular, the authors are conflating the analytical method ‘thematic synthesis’, with the type of evidence synthesis ‘qualitative evidence synthesis.’ Also, whilst I understand that it was not possible to register this review on PROSPERO, the lack of protocol (even if unpublished) is concerning. There are also some inaccuracies in the reporting and the review question remains unclear – this lack of clarity surrounding the review question at this stage, to me has implications on the quality of the review. Please see some more specific comments below: Abstract: In the design section, it would be helpful to include details of your eligibility criteria rather than just referring to the PerSPEcTiF framework. The sentence ‘Drawing upon the principles of thematic synthesis, a qualitative synthesis was performed based on line-by-line inductive coding of included studies’ needs rephrasing as qualitative synthesis is the type of evidence synthesis not the analysis method. I would suggest just stating here that included studies were analysed using thematic synthesis. As per my previous review, it would also be helpful to include details of whether screening, data extraction and quality appraisal were undertaken independently by two reviewers. Usually, the number of included studies would be included in the results section of the abstract. The text that has been added to the conclusion of the abstract is unclear to me. You refer to a juxtaposition of ESW perceptions and organisational priorities but on reading the results section of the abstract it is unclear what these are or what you mean by this. This needs to be much clearer for a reader who may only read the abstract. I think this is in part, because the results section has not gone beyond listing themes that you identified. Please can you provide some greater interpretation and explanation of your findings to help reader interpretation and understanding of your headline findings. Introduction: The sentence starting ‘Psychotherapy in the form of single-session’ needs revising for readability. Apologies I did not spot this in my previous review. However this paragraph ‘This review was initiated in collaboration with key ambulance management stakeholders who expressed a need for research to guide decisions about wellbeing interventions for frontline staff, therefore the focus is on ambulance trusts, although the findings will also be of relevance to fire and police organisations’ would be better placed in the discussion and is something that the authors may reflect on in terms of the potential applicability of their findings. It also needs to be clearer what you mean by ‘focus on ambulance trust’ as your inclusion criteria suggest that you included any literature on ESWs.
--

	I understand that the review was conducted as part of an undergraduate student project, and so a PROSPERO registration was not possible. However, the uncertainty surrounding the research question at this stage is concerning as this will have implications on the robustness of the review methods. The authors have added in the term 'protection' . It is unclear to me what is meant by mental health protection - sometimes this is referred to as protection or recovery and sometimes protection and recovery. Although the review was not registered on PROSPERO, can the authors clarify whether a protocol was developed before initiating the review? The first paragraph of the methods is still unclear to me and unfortunately implies a lack of understanding surrounding the review methodology. Thematic synthesis is the analysis method and so should be discussed in an analysis section, it is not the review method, which is qualitative evidence synthesis. Thank you for adding the PerSPEcTiF framework. However, the Phenomenon of interest seems inconsistent with the review question. Thank you for adding a note about COVID, however my concern remains that the searches are now out of date and that including the literature around COVID would be important. Whilst I agree that this is a unique event, it is one which is likely to have lasting impacts on the health service and have changed the support available and the way in which people view and access this support. There are some inconsistencies and inaccuracies in the reporting of the methods. Please include search dates. I.e. from inception to March 2020. When describing who undertook screening the authors state that two reviewers independently reviewed full texts However in the strengths and limitations section you state 'Only one reviewer performed the initial screen of studies, possible allowing for selection bias, however a second reviewer was recruited to independently assess potentially relevant studies against eligibility criteria' Please can you state who was involved at each stage of screening (title, abstract, full text). Results: In turn, these are grouped into one of the following two overarching constructs: 'factors contributing to mental health protection' or 'factors influencing mental health help-seeking behaviour'. It would be helpful to describe what you mean by mental health protection and factors influencing health-seeking behaviour here as protection feels different to recovery to me – this is not my area and so apologies for my ignorance, but I think it needs to be more clearly defined throughout the manuscript. The theme 'organisational' is unclear. Please can you revise this theme to make it clearer 'at a glance' what you mean here. For instance, 'stigma as a help-seeking barrier' is much clearer. The patient and public involvement section would be better placed in the methods section. Further justification is required here to explain why there was no PPI involvement. Discussion There is no strengths and limitations section in the discussion
--	--

REVIEWER	Case, Rosalind Monash University, Department of Epidemiology and Preventive Medicine
REVIEW RETURNED	14-Jul-2021

GENERAL COMMENTS	Thank you for addressing these comments comprehensively.
--

VERSION 2 – AUTHOR RESPONSE

Manuscript ID bmjopen-2020-047814.R1 "The barriers and facilitators to help-seeking and psychological protection of trauma-exposed emergency service staff: A qualitative evidence synthesis."

Thank you for submitting your revised version of this manuscript and for considering my revisions. Unfortunately, I still have some concerns surrounding the robustness of this review. In particular, the authors are conflating the analytical method 'thematic synthesis', with the type of evidence synthesis 'qualitative evidence synthesis.'

We have restructured and added to the methodology section to highlight that the type of evidence synthesis used is qualitative evidence synthesis. In response to a further comment below, discussion of the analytical method, thematic synthesis, has now been moved from the methodology to the data synthesis section.

There are also some inaccuracies in the reporting and the review question remains unclear – this lack of clarity surrounding the review question at this stage, to me has implications on the quality of the review.

We have reworded the review question in the introduction for increased clarity. For the reviewer's reference the review aims are:

- to identify factors and contexts that may contribute to mental health and recovery from psychological difficulties experienced by emergency service workers (ESWs) exposed to occupational trauma
- to identify barriers and facilitators to help-seeking behaviour among trauma-exposed ESWs.

To identify inaccuracies, the entire manuscript has been re-read by all authors, with diffuse formatting changes made throughout in order to remove inaccuracies and enhance the reader's experience.

Abstract:

In the design section, it would be helpful to include details of your eligibility criteria rather than just referring to the PerSPEcTiF framework.

We have added details of eligibility criteria in the design section of the abstract.

The sentence 'Drawing upon the principles of thematic synthesis, a qualitative synthesis was performed based on line-by-line inductive coding of included studies' needs rephrasing as qualitative synthesis is the type of evidence synthesis not the analysis method. I would suggest just stating here that included studies were analysed using thematic synthesis.

We thank the reviewer for this suggestion, and have made this alteration to the sentence.

As per my previous review, it would also be helpful to include details of whether screening, data extraction and quality appraisal were undertaken independently by two reviewers.

While we agree that this extra information would be helpful, we have already exhausted the abstract word limit. In response to a later comment, we have reworded the literature screening section in the methods to clarify which authors were involved in each of the steps mentioned.

Usually, the number of included studies would be included in the results section of the abstract.

This sentence has been moved to the results section of the abstract.

The text that has been added to the conclusion of the abstract is unclear to me. You refer to a juxtaposition of ESW perceptions and organisational priorities but on reading the results section of the abstract it is unclear what these are or what you mean by this. This needs to be much clearer for a reader who may only read the abstract. I think this is in part, because the results section has not gone beyond listing themes that you identified. Please can you provide some greater interpretation and explanation of your findings to help reader interpretation and understanding of your headline findings.

The sentence in question has been reworded in order to more accurately reflect our findings, and now offers a greater level of interpretation.

Introduction:

The sentence starting 'Psychotherapy in the form of single-session' needs revising for readability.

This has been revised for greater readability.

Apologies I did not spot this in my previous review. However this paragraph 'This review was initiated in collaboration with key ambulance management stakeholders who expressed a need for research to guide decisions about wellbeing interventions for frontline staff, therefore the focus is on ambulance trusts, although the findings will also be of relevance to fire and police organisations' would be better

placed in the discussion and is something that the authors may reflect on in terms of the potential applicability of their findings. It also needs to be clearer what you mean by 'focus on ambulance trust' as your inclusion criteria suggest that you included any literature on ESWs.

Thank you for bringing this sentence to our attention. This has now been moved to the discussion under the heading 'Application of review findings to organisational interventions, policy and research'. The sentence has been re-formatted to clarify that while ambulance management stakeholders were involved in initiating this review, its findings extend to fire and police organisations, and we have reflected on this in the strengths and limitations section. We have reflected on this in our strengths and limitations section.

The authors have added in the term 'protection'. It is unclear to me what is meant by mental health protection - sometimes this is referred to as protection or recovery and sometimes protection and recovery.

The term protection was added in response to comments made by reviewer 2, in order to clarify that not all trauma-exposed individuals develop a mental health disorder. While some individuals must 'recover' from psychological distress or other mental health conditions experienced as a result of trauma exposure, others may be 'protected' from these conditions by the presence of preventative factors. The use of both of these terms allows more accurate communication of the findings and interpretation; the distinction between these terms has been added to the introduction. We also thank the reviewer for identifying this inconsistency; instances where the terms protection and recovery were used together, have now been reworded to enhance clarity.

The first paragraph of the methods is still unclear to me and unfortunately implies a lack of understanding surrounding the review methodology. Thematic synthesis is the analysis method and so should be discussed in an analysis section, it is not the review method, which is qualitative evidence synthesis.

We thank the reviewer for highlighting that the phrasing of the analysis method in the first paragraph was potentially confusing. We have now reformatted the first paragraph of the methods to highlight that this review is a qualitative evidence synthesis. Discussion of the analysis method has now been confined to the data synthesis section of the methods.

Thank you for adding the PerSPEcTiF framework. However, the Phenomenon of interest seems inconsistent with the review question.

The phenomenon of interest has been reworded to more accurately reflect the review question

Thank you for adding a note about COVID, however my concern remains that the searches are now out of date and that including the literature around COVID would be important. Whilst I agree that this is a unique event, it is one which is likely to have lasting impacts on the health service and have changed the support available and the way in which people view and access this support.

Thank you for this comment. We do acknowledge that the contemporaneousness of the searches has been impacted by the fact that the review was conducted in the run up to the COVID pandemic, combined with the fact that the paper has now been in peer review for over 10 months. We do agree that further research into the mental wellbeing of emergency service workers specifically in the context of the Covid pandemic is of vital importance, particularly given the unique impact that working through the pandemic will have had on traumatisation and mental wellbeing. We believe, however, that reporting the findings of this review is important, and that the conclusions do stand in their own right as broadly applicable to the phenomenon of interest. Rather than being out of date or irrelevant as such, we argue that we are able to report an important milestone in the literature with respect to the collective understanding of traumatisation in ESWs; an understanding that can be built on in further work. We accept the specific limitation of this review with respect to the pan-Covid literature, and have directly addressed this as a limitation in the “strengths and limitations” section of the paper.

There are some inconsistencies and inaccuracies in the reporting of the methods.

Please include search dates. I.e. from inception to March 2020.

We thank the reviewer for highlighting this inaccuracy, which has now been amended. Inaccuracies in the ENTREQ statement have also been corrected.

When describing who undertook screening the authors state that two reviewers independently reviewed full texts. However in the strengths and limitations section you state ‘Only one reviewer performed the initial screen of studies, possible allowing for selection bias, however a second reviewer was recruited to independently assess potentially relevant studies against eligibility criteria’ Please can you state who was involved at each stage of screening (title, abstract, full text).

The literature screening stage of the review has been reworded to clarify which author(s) were involved at each stage.

Results:

In turn, these are grouped into one of the following two overarching constructs: ‘factors contributing to mental health protection’ or ‘factors influencing mental health help-seeking behaviour’. It would be helpful to describe what you mean by mental health protection and factors influencing health-seeking behaviour here as protection feels different to recovery to me – this is not my area and so apologies for my ignorance, but I think it needs to be more clearly defined throughout the manuscript.

The distinction between the terms protection and recovery has now been made in the introduction. Sections in the manuscript where the term protection was used have now been reworded for increased clarity. To avoid confusion, the construct name has been changed to 'factors contributing to mental health'.

The theme 'organisational' is unclear. Please can you revise this theme to make it clearer 'at a glance' what you mean here. For instance, 'stigma as a help-seeking barrier' is much clearer.

The theme has been renamed 'organisational sources of support'.

The patient and public involvement section would be better placed in the methods section. Further justification is required here to explain why there was no PPI involvement.

We apologise for the incompleteness of the earlier PPI statement, and have now expanded it to justify our decision relating to PPI.

Discussion

There is no strengths and limitations section in the discussion

We apologise for the earlier omission of the strengths and limitations section in this part of the manuscript, this has now been added.

Also, whilst I understand that it was not possible to register this review on PROSPERO, the lack of protocol (even if unpublished) is concerning. Although the review was not registered on PROSPERO, can the authors clarify whether a protocol was developed before initiating the review? I understand that the review was conducted as part of an undergraduate student project, and so a PROSPERO registration was not possible. However, the uncertainty surrounding the research question at this stage is concerning as this will have implications on the robustness of the review methods.

We hope that adding clarity to the review questions and making our methodology more explicit and transparent will address this concern. I was required to submit a protocol as part of my assessment I can confirm a protocol was in place when this review was undertaken.

Reviewer: 2

Dr. Rosalind Case, Monsh University

Comments to the Author:

Thank you for addressing these comments comprehensively.

We are grateful for your comments which helped to improve the manuscript quality.

Reviewer: 1

Competing interests of Reviewer: None declared

Reviewer: 2

Competing interests of Reviewer: None

VERSION 3 – REVIEW

REVIEWER	Scantlebury, Arabella University of York, York Trials Unit, Department of Health Sciences
REVIEW RETURNED	04-Nov-2021
GENERAL COMMENTS	Thank you for addressing my comments so comprehensively.